# Calibration for Long-tailed Scene Graph Generation

## ABSTRACT

Miscalibrated models tend to be unreliable and insecure for downstream applications. In this work, we attempt to highlight and remedy miscalibration in current scene graph generation (SGG) models, which has been overlooked by previous works. We discover that obtaining well-calibrated models for SGG is more challenging than conventional calibration settings, as long-tailed SGG training data exacerbates miscalibration with overconfidence in head classes and underconfidence in tail classes. We further analyze which components are explicitly impacted by the long-tailed data during optimization, thereby exacerbating miscalibration and unbalanced learning, including: **biased parameters**, **deviated boundaries**, and **distorted target distribution**. To address the above issues, we propose the **C**ompositional **O**ptimization **C**alibration (**COC**) method, comprising three modules: i. A parameter calibration module that utilizes a hyperspherical classifier to eliminate the bias introduced by biased parameters. ii. A boundary calibration module that disperses features of majority classes to consolidate the decision boundaries of minority classes and mitigate deviated boundaries. iii. A target distribution calibration module that addresses distorted target distribution, leverages within-triplet prior to guide confidence-aware and label-aware target calibration, and applies curriculum regulation to constrain learning focus from easy to hard classes. Extensive evaluation on popular benchmarks demonstrates the effectiveness of our proposed method in improving model calibration and resolving unbalanced learning for long-tailed SGG. Finally, our proposed method performs best on model calibration compared to different types of calibration methods and achieves state-of-the-art trade-off performance on balanced learning for SGG. The source codes and models will be available upon acceptance.

## CCS CONCEPTS

• **Computing methodologies → Scene understanding**.

## KEYWORDS

Calibration, Long-tailed Scene Graph Generation, Parameter Calibration, Boundary Calibration, Target Distribution Calibration.

## 1 INTRODUCTION

Deep neural networks have been extensively applied across a diverse range of domains. However, despite their prominent success, recent studies have found that they are not well-calibrated [15]. In other words, they cannot ensure that prediction confidences reflect

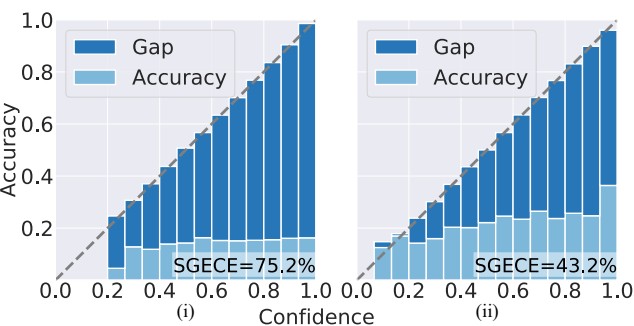

**Figure 1: Reliability diagrams [6] of Motifs model [62] on (i) original unbalanced and (ii) resampled more balanced [31] training data. Smaller gaps denote better calibration (i.e., less SGECE), so calibration is worse on unbalanced training data.**

the actual class probabilities. This gives rise to higher uncertainty and diminishing reliability in the decision-making process, posing safety risks when applied to safety-critical applications such as autonomous driving [14] and medical diagnosis [11].

Model calibration has been studied in the domains of classification [15, 26, 37, 65], detection [23, 27, 44], and segmentation [22, 52]. However, calibration for SGG remains under-explored. The primary objective of SGG is to convert a visual scene into a visually-grounded graphical representation [21], which has gained widespread utilization in safety-critical areas, e.g., autonomous driving [54, 61], medical diagnosis [17, 45], and robotics [1, 47], where reliability is crucial in real-life deployment. For instance, if SGG networks applied to autonomous driving cannot confidently predict the state of "human standing on the road", it will lead to unpredictable actions with worrisome consequences. Hence, it is essential for SGG networks not only to accurately represent structured scenes but also to be well-calibrated with high reliability.

Similar to conclusions drawn in other domains, current SGG models also exhibit poor calibration (c.f. Fig 1-left). However, we observe that calibration for SGG faces a unique challenge, where the unbalanced SGG training data leads to more miscalibrated models than more balanced data (c.f. Fig.1), characterized by overconfidence in head classes and underconfidence in tail classes (c.f. green lines in Fig.2-left). Additionally, such imbalanced data distribution causes a significant recall gap between the head and tail groups (c.f. hR@100 and tR@100 in Fig. 2-left). Although various works have proposed solutions to resolve the unbalanced learning in SGG [30, 36, 48, 63], they fail to reveal the miscalibration issue and evaluate their model calibration, limiting the reliability of their methods. Therefore, it is imperative to develop SGG models that insinuate both balanced learning and great calibration. To our knowledge, we are the first to explore calibration for long-tailed SGG, where we design a metric to measure calibration and propose a novel optimization calibration method to improve calibration and address unbalanced learning.

Current calibration methods can be categorized into two sets: post-hoc [15, 26, 37, 51] and train-time calibration methods [33, 65]. However, post-hoc methods primarily manipulate the posterior

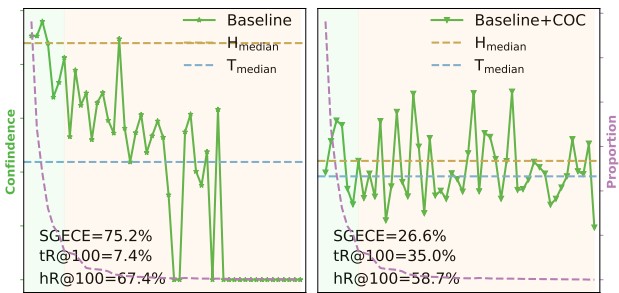

Figure 2: Distribution of predicted confidences. The left vertical axis (in green) and the right vertical axis (in purple) denote the confidence level and the size proportion of each class. Light green and light red spans indicate the head and tail groups. $H_{median}$ and $T_{median}$ denote the median confidences of head and tail classes. hR and tR are the mean recall of head and tail classes. Baseline exhibits severe overconfidence in head classes and underconfidence in tail classes, and a significant recall gap between head and tail classes. Our COC alleviates biased confidence and reduces the recall gap.

probability distribution without handling biased predictions. Instead, we focus on train-time methods to alleviate miscalibration and unbalanced learning. We identify three components that directly or indirectly influence confidence levels during optimization: parameters in the linear classifier, decision boundaries, and training target distribution. We present potential issues within these components that exacerbate miscalibration and unbalanced learning:

1) **Biased parameters.** As shown in Fig. 3 (a), the norm of both the weight and bias parameter in the linear classifier exhibits a biased distribution. The biased parameters amplify the discrepancy in confidence levels between the head and tail classes. 2) **Deviated boundaries.** Since majority classes are dense and minority classes are sparse, the feature extractor tends to map feature representations of the majority classes into dense clusters, whereas assigning minority classes to relatively sparse clusters [28, 69, 70]. This propensity easily leads to high feature deviation in minority classes and brings them to low-confidence regions [58]. Therefore, the feature points of the minority class are more likely to approach or even wrongly cross the decision boundaries with the majority class cluster (c.f. Fig. 3 (b)). 3) **Distorted target distribution.** SGG encodes the joint distribution $P(S,R,O)=P(S)P(O)P(R|S,O)\approx P(R|S,O)$ (we omit $(S,O)$ as they can be obtained from the pre-trained detectors [48, 49, 62]). Nonetheless, previous works [48, 49, 62] only capture the distribution of $P(R)$ while neglecting the within-triplet distribution $P(R|S,O)$. However, the relation set $R$ is long-tail distributed in SGGs [2, 48], leading models trained with such targets to be biased towards head classes, despite some tail classes having a higher probability of appearing in certain triplets (c.f. Fig. 3 (d)).

Subsequently, we propose a compositional calibration method to tackle the above three issues during optimization, which contain: 1) **P**arameter **C**alibration (**PC**) utilizes a hypersphere-based classifier that is invariant to the weight magnitude and removes the bias, eliminating the effects of biased parameters. 2) **B**oundary Calibration (**BC**) employs a dispersion loss to separate majority class features that consolidate the decision boundaries for minority classes to decrease their feature deviation (c.f. Fig. 3 (c)). 3) **T**arget

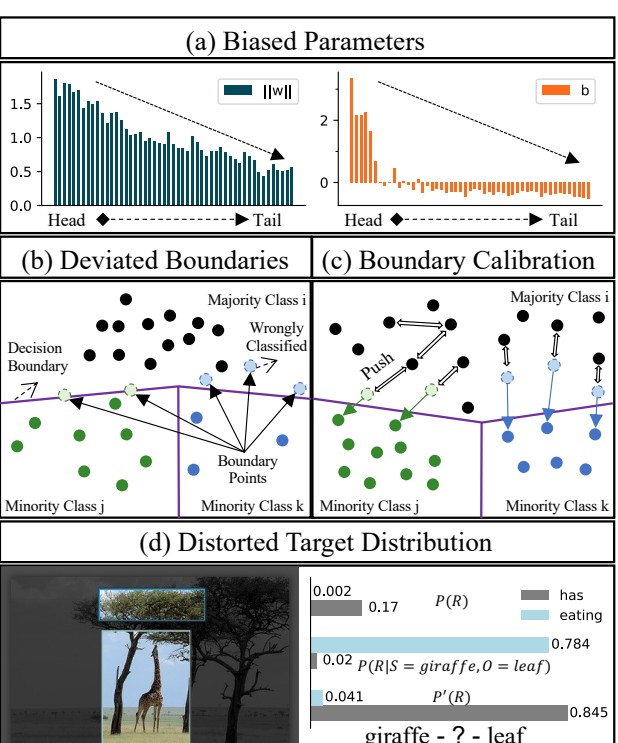

Figure 3: Three causes for miscalibration and unbalanced learning. (a) Biased parameters. Head classes have consistently larger $||w||$ and $b$ than tail classes. (b) Deviated boundaries. Minority classes exhibit high feature deviation, so they are prone to becoming close and even wrongly crossing the decision boundaries. (c) Boundary Calibration. Our method increases the separation between majority samples, thereby compressing minority samples into tighter clusters and consolidating the decision boundaries to reduce the feature deviation of minority classes. (d) Distorted target distribution. Given a triplet (giraffe,'relation', leaf), the head class 'has' dominates the relation distribution (top row), while the tail class 'eating' dominates the within-triplet prior (middle row). Baseline model [62] that learns through relation distribution $P(R)$ wrongly predicts the relation as 'has' (bottom row) with higher confidence, so the distorted target distribution needs calibration based on within-triplet distribution $P(R|S,O)$.

**D**istribution **C**alibration (**TDC**) leverages within-triplet prior to guide calibrating distorted target distribution in a confidence- and label-aware manner, and it incorporates a curriculum-based target regulation strategy to avoid excessive calibration. As seen in Fig. 2-right, our method is extremely effective in improving calibration (i.e., less SGECE) and balancing the miscalibrated confidence.

Our contributions include: 1) the first systematic study on calibration for long-tailed SGG with novel metrics and benchmarks (including qualitative and quantitative comparisons). 2) Identification of three issues exacerbating miscalibration and unbalanced learning, and introduction of a plug-and-play compositional calibration method to address them. 3) Extensive empirical results on SGG benchmarks using multiple baselines demonstrate the effectiveness of our method for calibration and achieving balanced learning.

## 2 RELATED WORKS

### 2.1 Long-tailed Scene Graph Generation

Many studies have attempted to address the long-tailed SGG problem. These works can be divided into three groups: 1) Re-sampling training data [7, 16, 31] or assigning different loss weights [32, 36, 59, 66, 67] to encourage balanced learning. 2) Generating additional labels [30, 60, 63] or features [29] to add training samples for tail classes. 3) De-biasing the biased probability distribution [2, 3, 39, 48]. However, these works overlook the miscalibration issue and its reasons, while we thoroughly analyze the core causes of miscalibration and present calibration methods to solve them.

### 2.2 Network Calibration

Well-calibrated models indicate an alignment between the predicted probabilities and true likelihood of correctness [13, 15, 43]. Model calibration has been explored in different tasks, including image classification [15, 26, 37, 65], semantic segmentation [8, 12, 22, 52], object detection [23, 27, 42, 44], medical imaging [38, 40], and so on. There are two main strategies to calibrate models: the first method re-scales posterior probabilities by applying parameters derived from the withheld portion of the training set during inference [20, 26, 37, 51]; the second strategy applies various techniques during training: Zhong et al. [65] utilize label smoothing to reduce the overconfidence; Mukhot et al. [41] argue that focal loss [33] calibrates models more effectively than cross-entropy loss; Shrivastava et al. [44] focus on detection and jointly calibrate multi-class confidence and box localization by leveraging their predicted uncertainties. However, the calibration of SGG models, which closely relates to real-world applications [1, 17, 45, 47, 54, 61], is rather overlooked. Current calibration methods neglect specific long-tailed challenges of SGG, so their applicability is limited in SGG.

## 3 METHOD

### 3.1 Preliminary

**Long-tailed SGG.** Given an image $I$, the objective of SGG is to generate a set of triplets $Tri = \{(s_i, o_i, r_i)\}_{i \in I}$ [21, 35, 68], where $r_i \in \mathcal{R}$ is the relationship, $s_i \in O$ is a subject, and $o_i \in O$ is an object. Each $s_i$ or $o_i$ consists of the bounding box $bb_i \in \mathbb{R}^4$ and object label $c_i$ obtained through a pre-trained detector [62] (e.g., Faster-RCNN [46]). As discussed in Sec. 1, training data of SGG is inherently long-tailed [48], where head classes (denoted by $\mathcal{H}$) occur significantly more frequently than tail classes (denoted by $\mathcal{T}$), which exacerbates miscalibration and unbalanced learning.

**Calibration for Long-tailed SGG.** For classification models, given the training data input $x \in \mathcal{X}$ and the corresponding label $y \in \mathcal{Y}$, the predicted outputs are class labels $\hat{y}$ with confidence scores $\hat{p}$. A perfectly calibrated classification model [15] satisfies this equation:

$$\underbrace{\mathbb{P}(\hat{y} = y | \hat{p} = p)}_{\text{accuracy given } y, p} = \underbrace{p}_{\text{confidence}} \quad \forall p \in [0, 1], \quad (1)$$

where $p$ is the expected confidence, and $\mathbb{P}(\hat{y} = y | \hat{p} = p)$ is the predicted accuracy. A perfectly calibrated model should maintain close consistency between its accuracy and confidence scores [15, 52]. Expanding on this concept, we discuss model calibration for SGG. In SGG tasks (including PredCls, SGCls, and SGDet; more

details are in Sec.4.1), the final predicted relation mainly depends on the relation posterior probability distribution [29, 30, 48, 49, 63], so we focus solely on the relation classification calibration and no longer consider the calibration of the object classification to ensure a fair comparison. For PredCls and SGCls tasks, the object pair $(s, o)$ can be known. Therefore, we select predictions with label $r$ whose object pair matches a unique ground truth object pair. A well-calibrated SGG model under these two settings should satisfy Eq. 1. Subsequently, we use the widely adopted metric Expected Calibration Error (ECE) [15, 27, 44] to measure model calibration:

$$\text{ECE} = \mathbb{E}_{\hat{p}} \left[ |\mathbb{P}(\hat{r} = r | \hat{p} = p) - p| \right]. \quad (2)$$

Equal binning on the confidence intervals is usually employed to estimate Eq. 2 [15] and can be formalized as follows:

$$\text{ECE}_{\mathcal{D}_b} = \sum_{b=1}^{B} \frac{|\mathcal{D}_b|}{|\mathcal{N}|} \left| \mathbf{acc}(\mathcal{D}_b) - \mathbf{conf}(\mathcal{D}_b) \right| \times 100\%, \quad (3)$$

where $B$ is the number of confidence bins with equal intervals, $\mathcal{D}_b$ is the sample set in bin $b$, $\mathcal{N}$ is the set of all samples, $\mathbf{conf}$ is the predicted confidence, and $\mathbf{acc}$ is the accuracy of all samples. However, in long-tailed SGG, head classes extremely outweigh the occurrence of tail classes, so the ECE metric may understate the calibration performance of tail classes. Specifically, for a typical tail class where both the accuracy and confidence levels are low, its contribution to the final overall becomes negligible despite poor performance. Hence, we devise a novel metric, MECE (**M**ean-accuracy **ECE**), to better accommodate the tail classes and gain a comprehensive understanding of the calibration of each class, given as:

$$\text{MECE}_{\mathcal{D}_b} = \sum_{b=1}^{B} \frac{|\mathcal{D}_b|}{|\mathcal{N}|} \left| \mathbf{macc}(\mathcal{D}_b) - \mathbf{conf}(\mathcal{D}_b) \right| \times 100\%, \quad (4)$$

where $\mathbf{macc}$ denotes averaging accuracy across each class.

In the SGDet task, the boxes of object pair $(s, o)$ cannot be known by default, so we only select object pairs that match the ground truth pairs. Specifically, the IOU (Intersection of Union) of selected object pairs should satisfy $(\text{IoU}(\hat{bb}_s, bb_s) > 0.5)\&(\text{IoU}(\hat{bb}_o, bb_o) > 0.5)$, and the predicted classes should satisfy $(\hat{c}_s = c_s)\&(\hat{c}_o = c_o)$. However, unlike in PredCls and SGCls tasks, for a selected triplet, there may not exist only a one-to-one correspondence to a single ground truth triplet, but rather a set of ground truth triplets with relation set $\{r_1, r_2, ...\}$. Therefore, in the SGDet task, we construct the new ground truth relation set $\mathcal{R}_{tri} = \bigcup_{n=1}^{N} \{r_1, r_2, ...\}_n$, and well-calibrated SGG models should adhere to this equation:

$$\text{ECE} = \mathbb{E}_{\hat{p}} \left[ |\mathbb{P}(\hat{r} = r_{tri} | \hat{p} = p) - p| \right] \quad \forall r_{tri} \in \mathcal{R}_{tri}. \quad (5)$$

Finally, we apply Eq. 4 to estimate the metric in Eq. 5 under the SGDet task. To simplify the description, we will refer to these two types of ECE for SGG as **SGECE** (**S**cene **G**raph **ECE**) in this paper.

### 3.2 Model Optimization for SGG

Current SGG models rely on a linear classifier to learn the relation classes, where the predicted logit of relation class $i$ is given by:

$$z_i = W_i^T f + b_i = ||W_i|| \cdot ||f|| \cos \theta_i + b_i, \quad (6)$$

where $z$ denotes the output class logit, $W$ denotes the weight of the classifier, $b$ represents the learned bias, and $f$ is the extracted relation feature by relation context models [49, 62]. $\theta_i$ is the angle

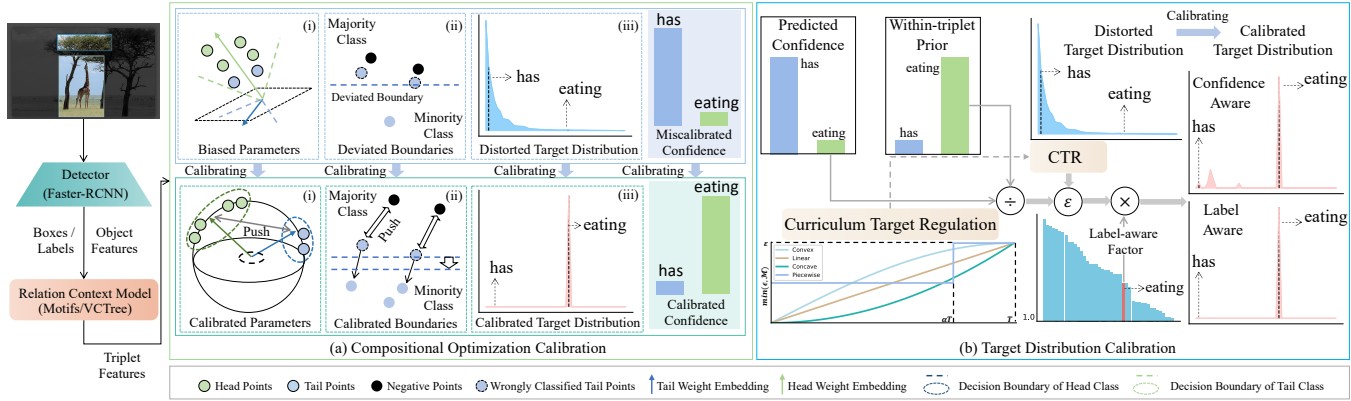

**Figure 4: (a) The pipeline of the compositional optimization calibration method. It consists of (i) parameter calibration, (ii) boundary calibration, and (iii) target distribution calibration to separately address biased parameters, deviated boundaries, and distorted target distribution issues. (b) The pipeline of the target distribution calibration method that calibrates the distorted relation target distribution. First, we use within-triplet prior and the predicted confidence to conclude the transferring factor $\epsilon$. Then, we utilize a label-aware factor to assign greater importance to the harder-to-learn tail classes. Finally, the curriculum target regulation module is applied to balance learning between easy and hard classes by regulating the size of $\epsilon$.**

between the class weight $W_i$ and feature $f$. Given the training target distribution $q$, the empirical loss function for SGG is as follows:

$$\mathbb{E}_{\boldsymbol{q}}[\mathcal{L}(\boldsymbol{z},\boldsymbol{y})] = \int_{\boldsymbol{q}} \boldsymbol{q}(\boldsymbol{x})\mathcal{L}(\boldsymbol{z},\boldsymbol{y})d\boldsymbol{x}. \tag{7}$$

The predicted confidence scores $\hat{\boldsymbol{p}}$ and classes $\hat{\boldsymbol{r}}$ are:

$$\hat{\boldsymbol{p}} = \max softmax(\boldsymbol{z}) \quad \hat{\boldsymbol{r}} = \underset{\{1,2,\cdots,\mathcal{R}\}}{\arg\max} \hat{\boldsymbol{p}}. \tag{8}$$

Based on Eq. 6-8, we argue that for a given relation feature, its predicted confidence and class mainly depend on: 1) magnitude of weight vector $W$ and bias $b$ in Eq. 6; 2) learned decision angle $\theta$ in Eq. 6; 3) the target distribution $q$ in Eq. 7. Motivated by these insights, to study miscalibration and unbalanced learning for long-tailed SGG, we explore underlying issues associated with these three elements (c.f. Sec. 1) and find: biased parameters, deviated boundaries, and distorted target distribution. Subsequently, we propose a compositional optimization calibration method (c.f. Fig. 4) including three specialized calibration methods to address the above issues: parameter calibration (Sec. 3.3.1), boundary calibration (Sec. 3.3.2), and target distribution calibration (Sec. 3.3.3).

## 3.3 Compositional Optimization Calibration

### 3.3.1 *Parameter Calibration.*
As discussed in Sec. 1, the magnitudes of weight and bias in the current classifier are biased, leading to unbalanced learning and exacerbated miscalibration. Specifically, as shown in Fig. 5 (a), decision boundaries tend to be closer to the minority class weight that has a smaller magnitude, which causes minority samples to be more easily far away from their own class centroids and obtain low confidences [24]. Based on these observations, we utilize a magnitude-invariant classifier defined on the unit hypersphere [10, 18, 53] to tackle the biased parameter problem.

**Triplet Feature.** To capture robust contextual information for the triplet feature $f$ prior to the classification stage, we fuse object, union, and spatial features in the following fashion:

$$f = \mathrm{MLP}((f_s * f_o * f_u) \oplus f_{spa}), \tag{9}$$

where $f_s$ and $f_o$ are feature vectors of the subject and object, respectively, captured by the object refined model [34, 49, 62]. $f_u$ is the feature for the union region occupied by the object pair. $f_{spa}$ is the bounding-box spatial feature for the object pair [34]. $\oplus$ denotes the feature concatenation. An MLP (Multi Layer Perceptron) is used to fuse these features to get the final refined triplet feature $f$ [64].

**Learning on Hypersphere.** For the classifier in SGG models, we optimize it on hypersphere; its predicted logit for the $i^{th}$ class is:

$$z_i = ||\boldsymbol{W}_i|| \cdot ||\boldsymbol{f}|| \cos\theta_i + \boldsymbol{b}_i = \tau\cos\theta_i$$
$$s.t. ||\boldsymbol{W}_i|| = 1, ||\boldsymbol{f}|| = \tau, \boldsymbol{b}_i = 0. \tag{10}$$

Specifically, we fix $||\boldsymbol{W}_i|| = 1$ by L2 normalization and set $\boldsymbol{b}_i = 0$. Moreover, we fix the feature $f$ by L2 normalization and re-scale it to $\tau$ as [53]. By doing so, classifier optimization is performed on the unit hypersphere, unaffected by the magnitudes of weight and bias.

**Discriminative Regularization.** We then incorporate a discriminative regularization on the weight vectors across all classes, aiming to scatter weights across the hypersphere and promote more discriminative boundaries between various classes [34, 64], given as:

$$\mathcal{L}_{reg} = \frac{1}{|\mathcal{R}|^2} \sum_{i=0}^{|\mathcal{R}|} \sqrt{\sum_{j=0}^{|\mathcal{R}|} (\overline{\boldsymbol{W}}_i^T \overline{\boldsymbol{W}}_j)^2}, \tag{11}$$

where $\bar{v} = v/||v||_2$ denotes the L2-normalized vector. As shown in Fig. 5 (b), this regularization can enlarge the inter-class angular distances of class weights, improving the inter-class separability.

### 3.3.2 *Boundary Calibration.*
Though parameter calibration can eliminate the impact of biased parameters, the sparse data of minority classes leads to high feature deviation [56, 58], while majority classes benefit from more compact feature vectors and exhibit lower feature deviation, so the actual decision boundaries may remain deviated rather than as ideal as depicted in Fig. 5 (b). As shown in Fig. 5 (c), given a minority sample of class $i$, the deviated boundary results in a larger angular distance between the sample and its own class weight than the weight of another majority class $j$ (i.e.,

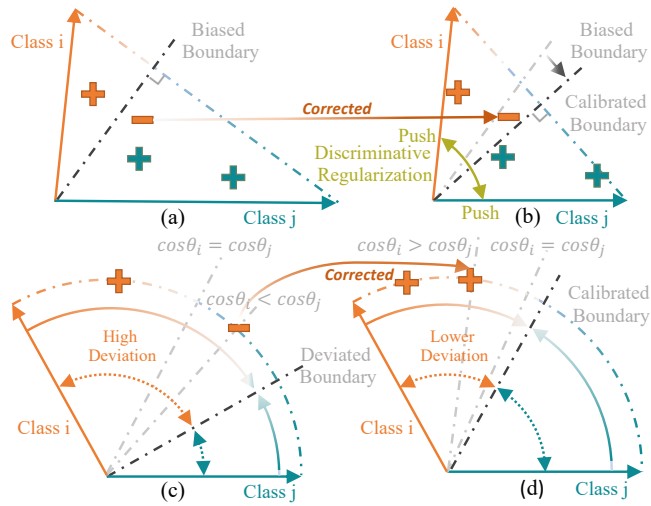

**Figure 5: (a)-(b) Illustration of the learning space of minority class $i$ and majority class $j$. The biased decision boundary is induced by imbalanced weight magnitudes. Transferring the weight optimization to the hypersphere alleviates the biased boundary. Discriminative regularization can widen the distance between inter-class centroids. (c)-(d). By calibrating the boundary of minority class $i$ with a high deviation to a lower deviation, the misclassified sample is correctly classified.**

$cos\theta_i < cos\theta_j$). This phenomenon causes the minority sample to be inclined toward low-confidence regions [58] and easily ill-classified.

We address the problem by dispersing the features of majority classes to consolidate the decision boundaries of minority classes. By scattering the features of the majority classes far from each other, it encourages the features of minority classes to become more compact, thereby decreasing their feature deviation. This approach enables tail samples to obtain higher decision confidences (i.e., $cos\theta_i > cos\theta_j$ in Fig.5 (d)) to alleviate miscalibration, which can also increase the likelihood of correct classification for them. Based on these insights, we present the dispersion loss in this section.

**Pairwise Dispersion.** To scatter the samples of majority classes from each other in the feature space, we start by quantifying the distance between them. A natural choice is the L2 distance, where $\delta(m, n) = \|m - n\|_2$. Given that negative samples appear much more frequently than head or tail samples, we choose to disperse their features. Moreover, the head classes already tend to exhibit overconfidence; their boundaries are less deviated compared to the tail classes, so we only calibrate the boundaries of the tail classes. However, we find it quickly becomes computationally unscalable if we attempt to push every negative sample away from each other. To address this, we relax the condition and only maximize the distance between features of negative triplets that share the same subject $s$ and object $o$. Specifically, given triplet features $f_i^{s,r_i,o}$ and $f_j^{s,r_j,o}$, where $r_i = r_j = c_n$ and $c_n$ is the negative class index, we apply the L2-normalized triplet features $\overline{f}$ as used in Eq. 10 and define the pairwise dispersion between selected samples as follows:

$$d_{i,j} = \frac{\delta(\overline{f}_i^{s,r_i,o}, \overline{f}_j^{s,r_j,o})}{\max(\delta(\overline{f}_i^{s,r_i,o}, \overline{f}_j^{s,r_j,o}))}. \tag{12}$$

**Dispersion Loss.** Following the above definition, we optimize the model by maximizing the dispersion between features of these sets of negative samples through the proposed dispersion loss:

$$\mathcal{L}_{neg}(x) = \frac{1}{|\mathcal{N}_{s,o}|} \sum_{x_{i,j} \in \mathcal{N}_{s,o}} e^{-d_{i,j}}, \tag{13}$$

where $\mathcal{N}_{s,o}$ denotes all sets of negative samples that share the same subject and object. This loss is influenced by distance $\delta_{i,j}$; the smaller the distance, the greater the contribution to the loss. Therefore, it can disperse the negative features within the same object pair and further reduce the feature deviation of tail classes.

*3.3.3* **Target Distribution Calibration.** Current SGG models do not take into account the within-triplet distribution given by the conditional distribution $P(R|S, O)$ as discussed in Sec. 1. Instead, simply fitting the distribution $P(R)$ makes the predicted probabilities biased towards head classes. This causes some tail class samples, which may be dominant in certain object pairs, to be underfit and have low predicted confidences. Our target distribution calibration method attempts to resolve distorted target distribution by calibrating the training target based on within-triplet distribution.

However, we observe that the triplet class amount is extremely large (e.g., about 26,300 in the VG dataset [25]), making it challenging to directly fit within-triplet distribution data. Therefore, we opt to transfer the target from the within-triplet distribution $q_t$ to the relation distribution $q_r$ and construct the empirical loss as:

$$\mathbb{E}_{q_t}[\mathcal{L}(z, r)] = \int_{q_t} q_t(x)\mathcal{L}(z, r)dx$$
$$= \int_{q_t} \frac{q_t(x)}{q_r(x)} q_r(x)\mathcal{L}(z, r)dx$$
$$= \int_{q_r} \frac{q_t(x)}{q_r(x)} \mathcal{L}(z, r)dx \tag{14}$$
$$= \mathbb{E}_{q_r}[\epsilon(x)\mathcal{L}(z, r)]$$
$$= \frac{1}{|\mathcal{N}|} \sum_{x \in \mathcal{N}} \epsilon(x)\mathcal{L}(z, r),$$

where $\epsilon$ is the transferring factor to transfer target distribution.
**Confidence-aware TDC (CTDC).** In SGG models, we further find the predicted relation distribution generated by the current model better reflects the extent of overconfidence or underconfidence compared to the statistical relation distribution derived from training data (c.f. an example in Fig. 3 (d)), so we apply the predicted relation distribution as $q_r$ in Eq. 14. Additionally, we define the empirical within-triplet prior $q_{s,o}$ for each set of $s$ and $o$ to constitute $q_t$ as:

$$q_{s,o} = P(R|S = s, O = o) = \frac{C_{s,r,o}}{\sum_{r'} C_{s,r',o}}, \tag{15}$$

where $C_{s,r,o}$ is the count of $(s, r, o)$ triplet, which is obtained from training data [2].We then define the transferring factor $\epsilon$ as:

$$\epsilon(x) = \frac{q_t(x)}{q_r(x)} = \frac{q_{s,o}(x)}{q_r(x)} = \begin{cases} \frac{q_{s,o}(x)}{q_r(x)}, & r \in \mathcal{T} \\ q_{s,o}(x), & r \in \mathcal{H}, \end{cases} \tag{16}$$

where $\epsilon$ can dynamically transfer the predicted relation distribution to align the current target with the within-triplet distribution. For head classes, $q_r(x)$ is set as 1, as they are overconfident and do not need further transferring targets. To ensure $\epsilon$ of tail classes exceeds

that of head classes, the lower bound of $\epsilon$ in tail classes is set to 1. Then, we use the following loss to optimize the calibrated targets:

$$\mathcal{L}_{tdc}(x) = \frac{1}{|\mathcal{N}|} \sum_{x \in \mathcal{N}} -\epsilon(x) \log \hat{p}. \quad (17)$$

**Label-aware TDC (LTDC).** Nevertheless, we find there is variation in distribution within tail classes, where some tail classes appear much more frequently than others, so it becomes increasingly difficult to learn transferred targets with a smaller number of available samples. Finally, the CTDC module may not always achieve the desired calibration effect. Therefore, we further propose a label-aware TDC on top of CTDC to deal with inter-tail-class distribution differences. Specifically, we take into account the sample size of tail classes [5, 65], and design a label-aware factor that increases with smaller class sizes, which can assist harder-to-learn tail classes to effectively arrive at the optimal objective, formulated as follows:

$$\epsilon(x) = \begin{cases} \frac{q_{s,o}(x)}{q_r(x)} * \frac{(1-\beta^{n(x)})}{(1-\beta^{n_{max}})}, & r \in \mathcal{T} \\ q_{s,o}(x), & r \in \mathcal{H}, \end{cases} \quad (18)$$

where $\beta$ is a hyperparameter between 0 and 1, $n$ is the sample size of tail classes, and $n_{max}$ is the maximum sample size in tail classes. To ensure the effectiveness of CTDC, the label-aware factor should not fall below 1, so we set $1-\beta^{n_{max}}$ as the denominator. In this manner, the label-aware factor assigned to the largest sample size remains 1, while progressively increasing with smaller sample sizes.

**Curriculum Target Regulation (CTR).** A well-designed balanced learning method should achieve an optimal trade-off across all classes [4, 29]. However, we find that the unrestrained target calibration mechanisms may excessively reduce the confidence of traditionally easy-to-learn head classes and impair their performance. We refer to this problem as overcalibration. Inspired by curriculum learning [55], we try different scheduling functions to regulate the learning focus from easy-to-learn head classes to hard-to-learn tail classes. Specifically, we consider modulating the transferring factor $\epsilon(x)$ by $\mathcal{H}(\epsilon(x), t)$, a curriculum function that is related to the current training iteration $t$. As shown in Fig. 4 (b), we first try three conventional single curriculum functions in TDC:

1) Linear form: $\mathcal{H}(\epsilon(x), t) = \frac{t}{T} * \epsilon(x)$;

2) Convex form: $\mathcal{H}(\epsilon(x), t) = sin(\frac{t\pi}{2T}) * \epsilon(x)$;

3) Concave form: $\mathcal{H}(\epsilon(x), t) = (\frac{t}{T})^2 * \epsilon(x)$.

Here, $T$ is the total number of iterations. In addition to the single curriculum functions, we introduce a piecewise curriculum function shown in Eq. 19. In the first stage, an upper limit $\mathcal{M}$ is used to actively restrict the excessive calibration of $\epsilon$ on tail classes to ensure the effective learning of easy-to-learn head classes. In the second stage, there are no longer constraints on the factor $\epsilon$, shifting more learning focus toward the hard-to-learn tail classes.

$$\mathcal{H}(\epsilon(x), t) = \begin{cases} min(\epsilon(x), \mathcal{M}), & t \le \alpha T, \alpha \in (0, 1) \\ \epsilon(x), & \alpha T < t \le T. \end{cases} \quad (19)$$

## 3.4 Training Objective

The overall training objective consists of the discriminative regularization loss (Eq. 11), dispersion loss (Eq. 13), and target distribution calibration loss (Eq. 17). The final loss function is:

$$\mathcal{L} = \mathcal{L}_{reg} + \mathcal{L}_{neg} + \mathcal{L}_{tdc}. \quad (20)$$

# 4 EXPERIMENT

## 4.1 Datasets and Metrics

**Datasets.** We conduct experiments on two large-scale, long-tailed SGG datasets: the Visual Genome (**VG**) [25] and **GQA-200** [19].

**Tasks.** We evaluate our proposed methods on three SGG tasks: 1) Predicate Classification (**PredCls**): The prediction of relation categories given object pairs' bounding boxes and their corresponding labels. 2) Scene Graph Classification (**SGCls**): The prediction of both object and relation categories given bounding boxes. 3) Scene Graph Generation (**SGDet**): The detection of bounding boxes and classification of both object pairs and their relationships.

**Metrics.** We evaluate our method from two aspects: effectiveness on calibration and balanced learning for SGG. In the former, we report the proposed metric **SGECE** to measure calibration. In the latter, we use three commonly-used metrics: Recall@K (**R@K**) [57, 62], mean Recall@K (**mR@K**) [31, 48], and the mean of the above two metrics, **MR@K**. R@K tends to be more biased towards head classes, while mR leans towards tail classes. MR@K provides a more comprehensive and balanced evaluation of all relation classes [30, 64]. More implementation details can be found in the appendix.

## 4.2 Effectiveness on Model Calibration

We first demonstrate the effectiveness of our method in addressing miscalibration, showcasing calibration performance both quantitatively and qualitatively. Specifically, we compare three types of methods: 1) specialized train-time calibration methods, e.g., **FL** (**F**ocal **L**oss) [41] and **LAS** (**L**abel-**A**ware **S**moothing) [65]; 2) probability post-processing method for long-tailed classification: **LA** (**L**ogit **A**djustment) [39, 50]; 3) SOTA balanced-learning methods for SGG, e.g., **FGPL** [36], **IETrans** [63] and **NICE** [30].

**Quantitative Analysis.** In Tab. 1, we have five observations: 1) The baseline model exhibits poor performance on SGECE, mR@K, and MR@K, indicating miscalibration and unbalanced learning. On the other hand, COC demonstrates superior performance on SGECE, mR@K, and MR@K, suggesting that COC effectively calibrates overconfidence in head classes and underconfidence in tail classes, resulting in substantial calibration improvement and more balanced recall performance. 2) COC achieves the best SGECE and MR@K, demonstrating its superior ability to mitigate miscalibration while achieving an optimal trade-off performance across all classes. 3) Specialized classification calibration methods (FL and LAS) obtain similar SGECE to other specialized balanced-learning methods in SGG, but their worse mR@K and MR@K performances indicate that they perform suboptimally on balanced learning for long-tailed SGG. This highlights the need for specialized calibration solutions tailored to long-tailed SGG. 4) LA is a subpar calibration method, as it performs worse than COC on nearly all metrics. 5) IETrans outperforms COC on mR@K, but its inferior SGECE reflects poorer calibration capability. The lower R@K demonstrates IETrans sacrifices much more majority class performance for the improvement of minority classes, so it is not an optimal calibration method.

**Qualitative Analysis.** As shown in Fig. 6, compared to other methods, COC can more effectively reduce the gaps between confidence and accuracy in reliability diagrams and achieve lower SGECE across all three SGG tasks, which proves the superior effectiveness of COC on model calibration for long-tailed SGG.

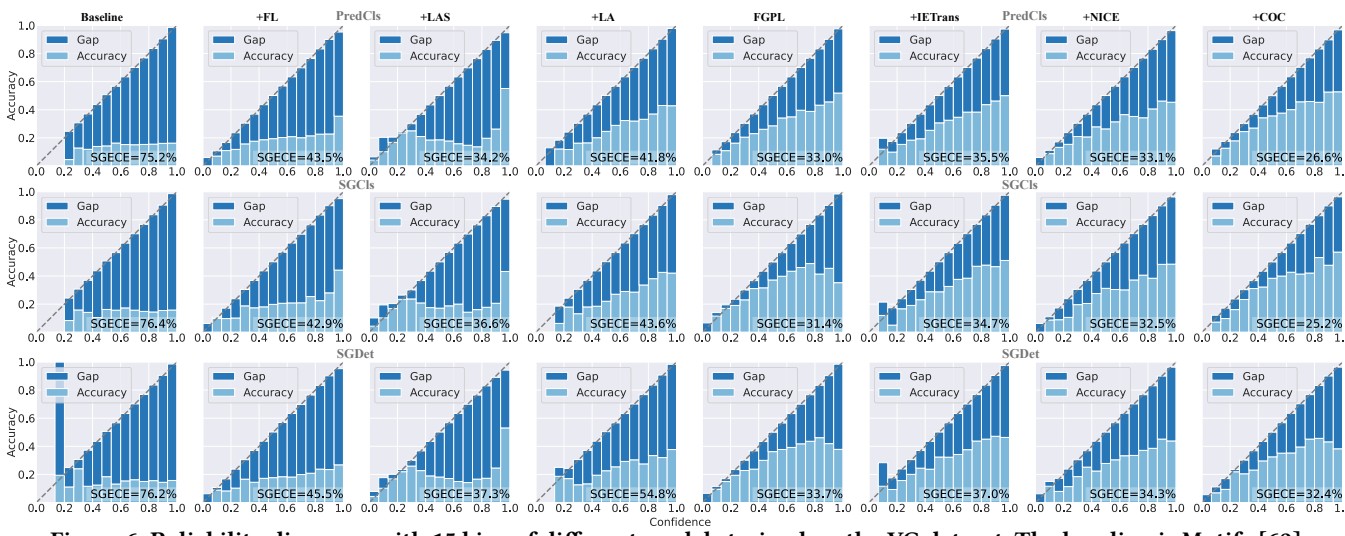

Figure 6: Reliability diagrams with 15 bins of different models trained on the VG dataset. The baseline is Motifs [62].

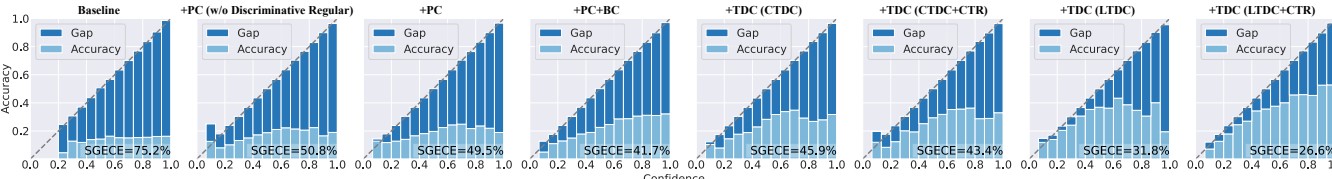

Figure 7: Reliability diagrams with 15 bins of each component in COC on the VG dataset (PredCls task).

| Models | PredCls | | SGCls | | SGDet | |
|--------|---------|-------|-------|-------|-------|-------|
| | R/mR/MR@100 | SGECE | R/mR/MR@100 | SGECE | R/mR/MR@100 | SGECE |
| Baseline [62] | 68.0 / 15.8 / 41.9 | 75.2 | 40.1 / 8.5 / 24.3 | 76.4 | 37.3 / 7.1 / 22.2 | 76.2 |
| +FL [41] | 66.7 / 18.6 / 42.7 | 43.5 | 40.6 / 10.4 / 25.5 | 42.9 | 33.7 / 8.2 / 21.0 | 45.5 |
| +LAS [65] | 65.7 / 19.2 / 42.5 | 34.2 | 40.1 / 10.6 / 25.4 | 36.6 | 35.6 / 8.5 / 22.1 | 37.3 |
| +LA [39] | 56.6 / 34.9 / 45.8 | 41.8 | 33.5 / 19.1 / 26.3 | 43.6 | 33.5 / 13.7 / 23.6 | 54.8 |
| +FGPL [36] | 55.3 / 36.8 / 46.1 | 33.0 | 28.1 / 21.6 / 24.9 | 31.4 | 27.9 / 17.4 / 22.7 | 33.7 |
| +IETrans [63] | 50.5 / **39.0** / 44.8 | 35.5 | 30.3 / **22.4** / 26.4 | 34.7 | 27.3 / **17.7** / 22.5 | 37.0 |
| +NICE [30] | 57.2 / 32.3 / 44.8 | 33.1 | 33.8 / 17.4 / 25.6 | 32.5 | 30.8 / 14.3 / 22.6 | 34.3 |
| **+COC** | 62.0 / 38.3 / **50.2** | **26.6** | 37.8 / 20.5 / **29.2** | **25.2** | 32.0 / 17.4 / **24.7** | 32.4 |

Table 1: Calibration performance of different models on the VG. We re-implement other works using their open-source projects on our platform to obtain the new SGECE results.

## 4.3 Effectiveness on Balanced Learning

Our method not only demonstrates strong effectiveness in model calibration but also in balanced learning for SGG. We compare it with SOTAs on long-tailed VG benchmarks. These methods can be divided into two groups.: 1) model-specific models, e.g., **BGNN** [31], **RU-NET** [34], and **PeNet** [64], 2) model-agnostic methods, e.g., **FGPL** [36], **IETrans** [63], **NICE** [30], Inf [2], **DKBL** [4] and **CFA** [29]. Our method belongs to the latter and exhibits good flexibility. We evaluate various baseline models to show the robustness and proficiency of our method, e.g., **Motifs** [62] and **VCTree** [49]. **VG.** We make four observations from Tab. 2: 1) Compared to the baselines, our method exhibits significant performance gains in mR@K and MR@K across three tasks. For example, it improves 114% ∼ 135% on mR@100 and 6% ∼ 20% on MR@100 across three tasks on the Motifs baseline. 2) Other methods that outperform ours in mR@K experience a more significant drop in R@K, indicating the excessive sacrifice of head-class performance for tail-class improvement. Conversely, COC enhances mR@K without substantial degradation in R@K and achieves superior MR@K, which shows that COC more effectively addresses underconfidence in tail classes

while maintaining balanced performance between various classes. 3) After applying RelNms to eliminate redundant predictions [34], our method surpasses other model-specific and model-agnostic approaches in mR@K and MR@K on Predcls and SGCls. 4) Although the mR@K performance of COC may not be optimal in the SGDet task, the lower drop in R@K and superior MR@K also indicate excellent balanced learning capabilities of COC in the SGDet task. **GQA-200.** We evaluate COC on GQA-200 (c.f. Tab. 3) and find it outperforms other SOTAs on MR@K across three tasks. We also confirm COC can improve model calibration on GQA-200 (c.f. appendix). These verify the scalability and generalizability of COC.

## 4.4 Ablation Study

**Effectiveness of Each Component.** In Fig. 7, we make three observations: 1) PC, BC, and the final TDC all decrease SGECE from the baseline, demonstrating their effectiveness in improving calibration. 2) The SGECE performance of CTDC is inferior to that of LTDC, demonstrating the necessity of focusing more on the harder-to-learn tail classes. 3) Without the CTR module, the SGECE of CTDC and LTDC slightly degrades. Inclusion of the CTR decreases SGECE, indicating its ability to enhance calibration. Furthermore, in Tab. 4, we observe that each module contributes to the improvement of mR@K and MR@K, indicating the validity of each module in addressing the unbalanced learning. Further analysis of each component in COC can be found in the appendix. **Influence of Components in TDC.** We conduct ablation studies on the components of TDC in Tab. 5. We obtain three observations: 1) The first row shows the effectiveness of CTDC in improving calibration and balanced learning, evident in the performance gains in SGECE, mR@K, and MR@K. 2) Comparing the first row with the

                                                                    

| Models | PredCls | | | SGCls | | | SGDet | | |
|---|---|---|---|---|---|---|---|---|---|
| | R@50/100 | mR@50/100 | MR@50/100 | R@50/100 | mR@50/100 | MR@50/100 | R@50/100 | mR@50/100 | MR@50/100 |
| BGNN [31] *CVPR '21* | 59.2 / 61.3 | 30.4 / 32.9 | 44.8 / 47.1 | 37.4 / 38.5 | 14.3 / 16.5 | 25.9 / 27.5 | 31.0 / 35.8 | 10.7 / 12.6 | 20.9 / 24.2 |
| RU-Net * [34] *CVPR '22* | 67.7 / 69.6 | - / 24.2 | - / 46.9 | 42.4 / 43.3 | - / 14.6 | - / 29.0 | 32.9 / 37.5 | - / 10.8 | - / 24.2 |
| PeNet-Rwt * [64] *CVPR '23* | 59.0 / 61.4 | 38.8 / 40.7 | 48.9 / 51.1 | 36.1 / 37.3 | 22.2 / 23.5 | 29.2 / 30.4 | 26.5 / 30.9 | 16.7 / 18.8 | 21.6 / 24.9 |
| Motifs [48, 62] *CVPR '18* | 65.3 / 67.2 | 14.9 / 16.3 | 40.1 / 41.8 | 38.9 / 39.8 | 8.3 / 8.8 | 23.6 / 24.3 | 32.1 / 36.8 | 6.6 / 7.9 | 19.4 / 22.4 |
| + FGPL [36] *CVPR '22* | 51.5 / 55.4 | 33.0 / 37.5 | 42.3 / 46.5 | 23.4 / 24.0 | 21.3 / 22.5 | 22.4 / 23.3 | 20.8 / 23.6 | **15.4 / 18.2** | 18.1 / 20.9 |
| + IETrans [63] *ECCV '22* | 48.6 / 50.5 | 35.8 / 39.0 | 42.2 / 44.8 | 29.4 / 30.2 | 21.5 / 22.8 | 25.5 / 26.5 | 23.5 / 27.2 | 15.5 / 18.0 | 19.5 / 22.6 |
| + NICE [30] *CVPR '22* | 55.1 / 57.2 | 29.9 / 32.3 | 42.5 / 44.8 | 33.1 / 34.0 | 16.6 / 17.9 | 24.9 / 26.0 | 27.8 / 31.8 | 12.2 / 14.4 | 20.0 / 23.1 |
| + Inf [2] *CVPR '23* | 51.5 / 55.1 | 24.7 / 30.7 | 38.1 / 42.9 | 32.2 / 33.8 | 14.5 / 17.4 | 23.4 / 25.6 | 23.9 / 27.1 | 9.4 / 11.7 | 16.7 / 19.4 |
| + DKBL [4] *ACMMM '23* | 54.2 / 55.6 | 35.4 / 37.6 | 44.8 / 46.6 | 31.3 / 31.9 | 20.4 / 21.4 | 25.9 / 26.7 | 25.8 / 29.3 | 14.1 / 16.7 | 20.0 / 23.0 |
| + CFA [29] *ICCV '23* | 54.1 / 56.6 | 35.7 / 38.2 | 44.9 / 47.4 | 34.9 / 36.1 | 17.0 / 18.4 | 26.0 / 27.3 | 27.4 / 31.8 | 13.2 / 15.5 | 20.3 / 23.7 |
| **+ COC** | 59.7 / 62.0 | 35.3 / 38.3 | 47.5 / 50.2 | 36.8 / 37.8 | 19.6 / 20.5 | 28.2 / 29.2 | 27.6 / 32.0 | 15.1 / 17.4 | **21.4 / 24.7** |
| **+ COC** * | 61.4 / 63.8 | **37.5 / 41.1** | **49.5 / 52.5** | 38.6 / 39.7 | **22.0 / 23.6** | **30.3 / 31.7** | - | - | - |
| VCTree [48, 49] *CVPR '19* | 65.4 / 67.2 | 16.7 / 18.2 | 41.1 / 42.7 | 46.7 / 47.6 | 11.8 / 12.5 | 29.3 / 30.1 | 31.9 / 36.2 | 7.4 / 8.7 | 19.7 / 22.5 |
| + FGPL [36] *CVPR '22* | 42.4 / 43.7 | **37.5 / 40.2** | 40.0 / 42.0 | 27.2 / 28.0 | **26.2 / 27.6** | 26.7 / 27.8 | 20.3 / 22.9 | **16.2 / 19.1** | 18.3 / 21.0 |
| + IETrans [63] *ECCV '22* | 48.0 / 49.9 | 37.0 / 39.7 | 42.5 / 44.8 | 30.0 / 30.9 | 19.9 / 21.8 | 25.0 / 26.4 | 23.6 / 27.8 | 12.0 / 14.9 | 17.8 / 21.4 |
| + NICE [30] *CVPR '22* | 55.0 / 56.9 | 30.7 / 33.0 | 42.9 / 45.0 | 37.8 / 39.0 | 19.9 / 21.3 | 28.9 / 30.2 | 27.0 / 30.8 | 11.9 / 14.1 | 19.5 / 22.5 |
| + Inf [2] *CVPR '23* | 59.5 / 61.0 | 28.1 / 30.7 | 40.1 / 41.8 | 40.7 / 41.6 | 17.3 / 19.4 | 29.0 / 30.5 | 27.7 / 30.1 | 10.4 / 11.9 | 19.1 / 21.0 |
| + DKBL [4] *ACMMM '23* | 53.8 / 55.2 | 35.6 / 37.6 | 44.7 / 46.4 | 31.8 / 32.5 | 19.8 / 21.4 | 25.8 / 27.0 | 25.7 / 29.3 | 13.2 / 15.7 | 19.5 / 22.5 |
| + CFA [29] *ICCV '23* | 54.7 / 57.5 | 34.5 / 37.2 | 44.6 / 47.4 | 42.4 / 43.5 | 19.1 / 20.8 | 30.8 / 32.2 | 27.1 / 31.2 | 13.1 / 15.5 | 20.1 / **23.4** |
| **+ COC** | 60.3 / 62.6 | 34.7 / 37.6 | 47.5 / 50.1 | 42.0 / 43.3 | 23.3 / 24.6 | 32.7 / 34.0 | 27.5 / 31.7 | 12.8 / 15.0 | **20.2 / 23.4** |
| **+ COC** * | 62.0 / 64.5 | 37.0 / **40.5** | **49.5 / 52.5** | 43.4 / 44.9 | 25.3 / **27.6** | **34.4 / 36.3** | - | - | - |

**Table 2: Performance of SOTA SGG models on VG. Models using the RelNms trick proposed in [34] are marked by ∗.**

| Models | PredCls | | | SGCls | | | SGDet | | |
|---|---|---|---|---|---|---|---|---|---|
| | R@50 / 100 | mR@50 / 100 | MR@50 / 100 | R@50 / 100 | mR@50 / 100 | MR@50 / 100 | R@50 / 100 | mR@50 / 100 | MR@50 / 100 |
| Motifs [9, 62] *CVPR '18* | 65.3 / 66.8 | 16.4 / 17.1 | 40.9 / 42.0 | 34.2 / 34.9 | 8.2 / 8.6 | 21.2 / 21.8 | 28.9 / 33.1 | 6.4 / 7.7 | 17.7 / 20.4 |
| + GCL [9] *CVPR '22* | 44.5 / 46.2 | **36.7 / 38.1** | 40.6 / 42.2 | 23.2 / 24.0 | 17.3 / 18.1 | 20.3 / 21.1 | 18.5 / 21.8 | **16.8 / 18.8** | 17.7 / 20.3 |
| + CFA [29] *CVPR '23* | - / - | 31.7 / 33.8 | - / - | - / - | 14.2 / 15.2 | - / - | - / - | 11.6 / 13.2 | - / - |
| **+ COC** | 56.1 / 57.9 | **36.7 / 38.7** | 46.4 / 48.3 | 28.4 / 29.1 | **18.2 / 19.0** | 23.3 / 24.1 | 23.9 / 27.5 | 15.4 / 17.8 | 19.7 / 22.7 |
| VCTree [9, 49] *CVPR '19* | 63.8 / 65.7 | 16.6 / 17.4 | 40.2 / 41.6 | 34.1 / 34.8 | 7.9 / 8.3 | 21.0 / 21.6 | 28.3 / 31.9 | 6.5 / 7.4 | 17.4 / 19.7 |
| + GCL [9] *CVPR '22* | 44.8 / 46.6 | 35.4 / 36.7 | 40.1 / 41.7 | 23.7 / 24.5 | 17.3 / 18.0 | 20.5 / 21.3 | 17.6 / 20.7 | **15.6 / 17.8** | 16.6 / 19.3 |
| + CFA [29] *CVPR '23* | - / - | 33.4 / 35.1 | - / - | - / - | 14.1 / 15.0 | - / - | - / - | 10.8 / 12.6 | - / - |
| **+ COC** | 55.8 / 57.7 | **37.3 / 39.4** | 46.6 / 48.6 | 27.2 / 27.8 | **17.9 / 18.8** | 22.6 / 23.3 | 22.3 / 25.6 | 13.3 / 15.3 | 17.8 / 20.5 |

**Table 3: Performance of SOTA SGG models on GQA-200.**

| M | Components | | | PredCls | | | |
|---|---|---|---|---|---|---|---|
| | PC | BC | TDC | R@50/100 | mR@50/100 | MR@50/100 | SGECE |
| 0 | ✗ | ✗ | ✗ | 65.2 / 67.2 | 14.9 / 16.3 | 40.1 / 41.8 | 75.2 |
| 1 | ✓ | ✗ | ✗ | 65.4 / 67.3 | 20.4 / 22.0 | 42.9 / 44.7 | 49.5 |
| 2 | ✓ | ✓ | ✗ | 64.1 / 66.1 | 24.7 / 26.0 | 44.4 / 46.1 | 41.7 |
| 3 | ✓ | ✗ | ✓ | 60.6 / 62.5 | 33.0 / 35.1 | 46.8 / 48.8 | 28.4 |
| 4 | ✓ | ✓ | ✓ | 59.7 / 62.0 | **35.3 / 38.3** | **47.5 / 50.2** | 26.6 |

**Table 4: The ablation study of each component in COC.**

| M | Components | | | PredCls | | | |
|---|---|---|---|---|---|---|---|
| | CTDC | LTDC | CTR | R@50/100 | mR@50/100 | MR@50/100 | SGECE |
| 0 | ✗ | ✗ | ✗ | 64.1 / 66.1 | 24.7 / 26.0 | 44.4 / 46.1 | 40.5 |
| 1 | ✓ | ✗ | ✗ | 63.2 / 65.6 | 25.5 / 28.2 | 44.4 / 46.9 | 45.9 |
| 2 | ✗ | ✓ | ✗ | 57.8 / 60.2 | 31.1 / 34.5 | 44.5 / 47.4 | 31.8 |
| 3 | ✓ | ✗ | ✓ | 63.3 / 65.5 | 27.9 / 30.4 | 45.6 / 48.0 | 43.4 |
| 4 | ✗ | ✓ | ✓ | 59.7 / 62.0 | **35.3 / 38.3** | **47.5 / 50.2** | 26.6 |

**Table 5: The ablation study of each component in TDC.**

| Curriculum | PredCls | | | |
|---|---|---|---|---|
| | R@50/100 | mR@50/100 | MR@50/100 | SGECE |
| TDC (w/o CTR) | 57.8 / 60.2 | 31.1 / 34.5 | 44.5 / 47.4 | 31.8 |
| w/ CTR-Concave | 59.6 / 61.5 | 33.9 / 36.8 | 46.8 / 49.2 | 26.9 |
| w/ CTR-Linear | 58.5 / 61.1 | 31.5 / 34.9 | 45.0 / 48.0 | 30.6 |
| w/ CTR-Convex | 58.1 / 60.3 | 31.5 / 35.3 | 44.8 / 47.8 | 30.9 |
| w/ CTR-Piecewise | 59.7 / 62.0 | **35.3 / 38.3** | **47.5 / 50.3** | **26.6** |

**Table 6: The ablation study of different curriculum functions.**

**Influence of Curriculum Function in TDC.** In Tab. 6, we experiment with different curriculums in TDC. We find: 1) Incorporating the target regulation improves both MR@K and SGECE. This indicates that CTR can adjust the learning process in a more balanced fashion and achieve better calibration performance. 2) Through empirical validation, the piecewise curriculum function performs the best, which demonstrates the validity of its design insights.

## 5 CONCLUSION

Calibration is essential yet neglected for SGG. We are the first to conduct a comprehensive study on calibration for long-tailed SGG. We analyze three factors that aggravate miscalibration and unbalanced learning. To solve them, we propose a compositional optimization calibration method. Specifically, our method comprises a parameter calibration module to eliminate biased parameters, a boundary calibration module to reduce deviated boundaries, and a target distribution calibration module to calibrate distorted target distribution. We also design a curriculum-based learning method to alleviate the overcalibration. Extensive experiments on SGG benchmarks validate the effectiveness of our method in improving calibration, with optimal trade-off performance across various metrics. Furthermore, our work may set the stage for future research on the SGG task to not only focus on evaluating long-tailed benchmarks but also on our established calibration benchmarks to ensure the reliability and secure application of their designed models.

second row reveals the efficacy of the label-aware strategy in boosting harder-to-learn tail classes, resulting in notable improvements in both mR@K, MR@K, and SGECE. 3) A comparison between the second and fourth rows reveals the effectiveness of CTR for addressing the overcalibration issue, thereby mitigating the drop in R@K and achieving better MR@K as well as improved calibration.

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
