# OpenReview forum: "Calibration for Long-tailed Scene Graph Generation"
_acmmm.org/ACMMM/2024/Conference — MM2024 Poster_

### Official Review · Reviewer_ykr3 · 2024-05-07

**Rating:** 4
**Confidence:** 3

**Summary:**

This paper introduces calibration into Scene Graph Generation, addressing the commonly encountered issue of imbalanced learning. The authors identified the primary causes of miscalibration as biased parameters, deviated boundaries, and distorted target distribution. To rectify these, the paper proposes three types of calibrations: parameter calibration through discriminative regularization, boundary calibration via dispersion loss, and target distribution calibration using confidence-aware and label-aware TDC with curriculum target regularization. The empirical study demonstrates improved calibration results using the proposed SGECE metric across VG and GQA-200 datasets for PredCls, SGCls, and SGDet tasks.

**Strengths:**

1. The paper is well-written and logically structured, effectively identifying and quantitatively analyzing the issues before proposing solutions.

2. It addresses the crucial issue of calibration in the context of imbalanced learning.

3. The figures are clear and facilitate understanding of each proposed component's issues and core ideas.

4. The effectiveness of the proposed method in terms of calibration is demonstrated across various datasets and baseline comparisons.

**Limitations:**

1. The paper relies exclusively on the proposed SGECE metric, which calculates mean accuracy minus confidence for each bin, for evaluating calibration effectiveness. This approach lacks a quantitative study and explanation of why SGECE is superior to ECE, as well as how the proposed method performs in terms of ECE. Clarifications on any observed differences and their potential reasons are needed.
2. Line 379 suggests that calibration is focused on classifier parameters (weight and bias), but this seems limited since the main imbalance stems from the input, which is also significantly related to feature extraction and designed loss. This makes parameter calibration on classifiers seem insufficient.
3. Line 498 raises concerns about ensuring prediction accuracy for minority classes that naturally tend to have low confidence due to underrepresentation in learning.
4. Does this paper assume the training and testing data share identical long-tailed distributions? If so. it is unrealistic and not ideal for model calibration considerations.
5. The method's non-posthoc nature should ideally lead to performance improvements in the main task; however, the proposed method underperforms in terms of R@50/100 across all baselines and datasets.
6. Appendix line 26 lacks information on the computational resources used; specifically, it does not state how many V100 GPUs were employed for the tasks.
7. The choice to set the number of bins in the reliability diagram to 15, as opposed to the common default of 10, should be justified, especially concerning its significance in calibration evaluation.

**Suitability:**

2

---

### Official Review · Reviewer_WFpZ · 2024-05-09

**Rating:** 4
**Confidence:** 1

**Summary:**

This paper addresses the miscalibration issue in scene graph generation (SGG) models, particularly in the context of long-tailed training data. It proposes the Compositional Optimization Calibration (COC) method, which comprises three modules to mitigate biases, consolidate decision boundaries, and address distorted target distributions. Extensive evaluations demonstrate the effectiveness of COC in improving model calibration and resolving unbalanced learning for long-tailed SGG, achieving state-of-the-art trade-off performance on balanced learning.

**Strengths:**

The analysis is comprehensive, the problem definition is clear, and the improvement in results is significant.

**Limitations:**

The x-axis of Figures 6 and 7 is not very clear.

The three issues mentioned in the abstract, biased parameters, deviated boundaries, and distorted target distribution, are not clearly defined.

The wording is a bit difficult to understand

**Suitability:**

3

---

### Official Review · Reviewer_Rd8u · 2024-05-25

**Rating:** 5
**Confidence:** 3

**Summary:**

The authors propose a novel calibration method for unbiased scene graph generation (SGG) to solve the mis-calibration between confidence and accuracy. The method tackles the mis-calibration problem in three directions: model parameters, classifier boundary and target distribution. The paper explains and reveals the above-mentioned issues existing in the current SGG model training, and proposes Compositional Optimization Calibration (COC) to solve them. Through extensive experiments, their proposed COC not only shows significant improvement in mis-calibration metrics such as ECE, but also achieves state-of-the-art mR/MR in the context of unbiased SGG.

**Strengths:**

- As far as I am concerned, confidence-accuracy mis-calibration is a novel direction to be considered in unbiased SGG. This is a good point, and the fact that the results of significant improvement in unbiased SGG metrics also make this paper more impressive.
- The proposed method achieves SOTA results in unbiased SGG is a significant contribution to the community.
- The paper explain the ideas in a clear way. The paper is organized and easy-to-follow.

**Limitations:**

- I would suggest that you compare the strategy to more unbiased SGG works. For instance, is there any similarity between your proposed classifier boundary calibration method and DLFE (Recovering the Unbiased Scene Graphs from the Biased Ones, ACMMM'21)? Comparing to NICE/NICEST (TPAMI'24)?

**Suitability:**

3

---

### Meta-Review · Area_Chair_gohf · 2024-06-28

**Recommendation:** Accept (Poster)
**Confidence:** 4

**Metareview:**

The reviewers agree that the paper is solid, novel, and well-written. They agree that the experiments are correct and show the superiority of their method. The authors have also addressed most of the reviewers' concerns appropriately. Therefore, I recommend accepting this article.